# Multi-level quantum noise spectroscopy

Youngkyu Sung [1,2✉], Antti Vepsäläinen [1], Jochen Braumüller[1], Fei Yan[1,5], Joel I-Jan Wang [1], Morten Kjaergaard [1,6], Roni Winik [1], Philip Krantz [1], Andreas Bengtsson [1], Alexander J. Melville[3], Bethany M. Niedzielski[3], Mollie E. Schwartz [3], David K. Kim[3], Jonilyn L. Yoder[3], Terry P. Orlando[1,2], Simon Gustavsson[1] & William D. Oliver [1,2,3,4✉]

System noise identification is crucial to the engineering of robust quantum systems. Although existing quantum noise spectroscopy (QNS) protocols measure an aggregate amount of noise affecting a quantum system, they generally cannot distinguish between the underlying processes that contribute to it. Here, we propose and experimentally validate a spin-locking-based QNS protocol that exploits the multi-level energy structure of a superconducting qubit to achieve two notable advances. First, our protocol extends the spectral range of weakly anharmonic qubit spectrometers beyond the present limitations set by their lack of strong anharmonicity. Second, the additional information gained from probing the higher-excited levels enables us to identify and distinguish contributions from different underlying noise mechanisms.

[1] Research Laboratory of Electronics, Massachusetts Institute of Technology, Cambridge, MA, USA. [2] Department of Electrical Engineering and Computer Science, Massachusetts Institute of Technology, Cambridge, MA, USA. [3] MIT Lincoln Laboratory, Lexington, MA, USA. [4] Department of Physics, Massachusetts Institute of Technology, Cambridge, MA, USA. [5] Present address: Shenzhen Institute for Quantum Science and Engineering, Southern University of Science and Technology, Shenzhen, Guangdong, China. [6] Present address: Niels Bohr Institute, University of Copenhagen, 2100 Copenhagen, Denmark. ✉email: youngkyu@mit.edu; william.oliver@mit.edu

tudying noise sources affecting quantum mechanical systems is of great importance to quantum information processing, quantum sensing applications, and the fundamental understanding of microscopic noise mechanisms[1–4]. Generally, a quantum two-level system—a qubit—is employed as a sensor of noise that arises from the qubit environment including both classical and quantum sources[2,5]. By driving the qubit with suitably designed external control fields and measuring its response in the presence of environmental noise, the spectral content of the noise can be extracted[6–10]. Such noise spectroscopy techniques are generally referred to as quantum noise spectroscopy (QNS) protocols. Over the past two decades, QNS protocols have been explored for both pulsed (free-evolution) and continuous (driven-evolution) control schemes and experimentally implemented across many qubit platforms—including diamond nitrogen vacancy centers[11,12], nuclear spins[6,13], cold atoms[14], superconducting quantum circuits[10,15–18], semiconductor quantum dots[19–22], and trapped ions[23]. Although these protocols have generally focused on Gaussian noise models, a new QNS protocol was recently developed and demonstrated that enables higher-order spectral estimation of non-Gaussian noise in quantum systems[24,25].

Since QNS protocols commonly presume a qubit platform, they have generally been developed within a two-level system approximation, without regard for higher energy levels. As a consequence, despite tremendous progress and successes, QNS protocols have certain limitations (for example, limited bandwidth) when applied to weakly anharmonic qubits such as the transmon[26,27], the gatemon[28,29], or the capacitively shunted flux qubit[30]. However, since weakly anharmonic superconducting qubits are among the most promising platforms being considered for realizing quantum information processors[31], noise spectroscopy techniques that incorporate the effects of higher-excited states in these qubits must be developed to further improve their coherence and gate performance.

Among existing QNS protocols, the spin-locking approach has been shown to be applicable to both classical and non-classical noise spectra. It is also experimentally advantageous, using a relatively straightforward relaxometry analysis to extract a spectral decomposition of the environmental noise affecting single qubits[16,32], and it has recently been extended to measure the cross-spectra of spatially correlated noise in multi-qubit systems[33]. As with many contemporary QNS protocols, the spin-locking approach presumes a two-level-system approximation. While this approximation holds at low frequencies, its validity breaks down as one attempts to perform noise spectroscopy at frequencies approaching and exceeding qubit anharmonicity (e.g., around 200–300 MHz is conventional superconducting transmon qubits) due to the impact of additional energy levels, leading to systematic errors in the extracted noise spectrum.

In this work, we develop a multi-level spin-locking QNS protocol and experimentally validate it using a flux-tunable transmon qubit and accounting for five energy levels. We demonstrate an accurate spectral reconstruction of engineered flux noise over a frequency range 50–300 MHz, overcoming the spectral limitations imposed by the sensor's relatively weak anharmonicity of ~200 MHz. Furthermore, by measuring the power spectra of dephasing noise acting on both the $|0\rangle$–$|1\rangle$ and $|1\rangle$–$|2\rangle$ transitions, we extract and uniquely identify noise contributions from both flux noise and photon shot noise, an attribution that is not possible within solely a two-level approximation.

## Results

### System Description.
We consider an externally-driven $d$-level quantum system ($d > 2$), which serves as the quantum noise sensor that evolves under the influence of its noisy environment (bath). Throughout this work, we consider only pure dephasing ($\sigma_z$-type) noise. The impact of energy relaxation ($T_1$) on our protocol is discussed in Supplementary Note 6. In the interaction picture with respect to the bath Hamiltonian $H_B$, the joint sensor-environment system can be described by the Hamiltonian:

$$H(t) = \hbar \sum_{j=1}^{d-1} \left[ \left( \omega_s^{(j)} + B^{(j)}(t) \right) |j\rangle\langle j| + \xi(t) \lambda^{(j-1,j)} \left( \sigma_+^{(j-1,j)} + \sigma_-^{(j-1,j)} \right) \right], \quad (1)$$

where $|j\rangle\langle j|$ is the projector for the $j$th level of the multi-level sensor. The sensor eigenenergies are $\hbar\omega_s^{(j)}$ with the ground state energy set to zero, and the $B^{(j)}(t)$ correspond to the time-dependent noise operators that longitudinally couple to the $j$th level of the sensor and cause level $j$ to fluctuate in energy. The raising and lowering operators of the sensor are denoted by $\sigma_+^{(j-1,j)} \equiv |j\rangle\langle j-1|$ and $\sigma_-^{(j-1,j)} \equiv |j-1\rangle\langle j|$, respectively. The external driving field is denoted by $\xi(t)$. We continuously drive the multi-level sensor with a signal

$$\xi(t) = A_{\text{drive}} \cos(\omega_{\text{drive}} t + \phi), \quad (2)$$

where $A_{\text{drive}}$, $\omega_{\text{drive}}$, and $\phi$ correspond to the amplitude, the frequency and the phase of the driving field, respectively, and we assume $\phi = 0$ without loss of generality. The parameter $\lambda^{(j-1,j)}$ represents the strength of the $|j-1\rangle$–$|j\rangle$ transition relative to the $|0\rangle$–$|1\rangle$ transition with $\lambda^{(0,1)} \equiv 1$.

When the drive frequency $\omega_{\text{drive}}$ is resonant with the $|0\rangle$–$|1\rangle$ transition frequency $\omega_s^{(0,1)}$ of the sensor, the first two levels form a pair of dressed states, $|+^{(0,1)}\rangle$ and $|-^{(0,1)}\rangle$. The level separation between dressed states is the Rabi frequency $\Omega^{(0,1)}$, and it is determined predominantly (although not exactly, as we describe below) by the effective driving strength $\lambda^{(0,1)} A_{\text{drive}} \equiv A_{\text{drive}}$[34,35]. These dressed states form the usual spin-locking basis $\{|+^{(0,1)}\rangle, |-^{(0,1)}\rangle\}$ of a conventional, driven two-level sensor[16,32,33].

We now generalize the two-level spin-locking concept to the case of a multi-level sensor. By resonantly driving at the frequency $\omega_s^{(j-1,j)} \equiv \omega_s^{(j)} - \omega_s^{(j-1)}$ of the transition between states $|j-1\rangle$ and $|j\rangle$, the system forms a pair of dressed states, $|+^{(j-1,j)}\rangle$ and $|-^{(j-1,j)}\rangle$, separated by a Rabi frequency $\Omega^{(j-1,j)}$ (see Fig. 1a) that is determined predominantly by an effective driving strength $\lambda^{(j-1,j)} A_{\text{drive}}$. The effective two-level system formed by the basis $\{|+^{(j-1,j)}\rangle, |-^{(j-1,j)}\rangle\}$ acts as the $j$th spectrometer and probes dephasing noise that leads to a fluctuation of the $|j-1\rangle$–$|j\rangle$ transition at frequency $\Omega^{(j-1,j)}$. The case $j=1$ then corresponds back to the conventional two-level noise sensor.

Throughout the main text, we will refer to the reference frame and two-dimensional subspace defined by the $j$th spin-locking basis $\{|+^{(j-1,j)}\rangle, |-^{(j-1,j)}\rangle\}$ as the $j$th *spin-locking frame* and the $j$th *spin-locking subspace*, respectively. To move to the $j$th spin-locking frame, we apply unitary transformations and truncate the Hilbert space of the multi-level sensor into the $j$th spin-locking subspace (see detailed derivation in Supplementary Note 4). Then, the effective Hamiltonian describing the $j$th noise spectrometer is:

$$\tilde{H}_{\text{SL}}^{(j-1,j)}(t) = \frac{\hbar}{2} \left[ \Omega^{(j-1,j)} + \tilde{B}_{\parallel}^{(j-1,j)}(t) \right] \tilde{\sigma}_z^{(j-1,j)} + \hbar \tilde{B}_{\perp}^{(j-1,j)}(t) \left( \tilde{\sigma}_+^{(j-1,j)} + \tilde{\sigma}_-^{(j-1,j)} \right), \quad (3)$$

where $\tilde{\sigma}_z^{(j-1,j)}$, $\tilde{\sigma}_+^{(j-1,j)}$, and $\tilde{\sigma}_-^{(j-1,j)}$ denote the Pauli Z operator, the raising operator, and the lowering operator of the $j$th spin-locked

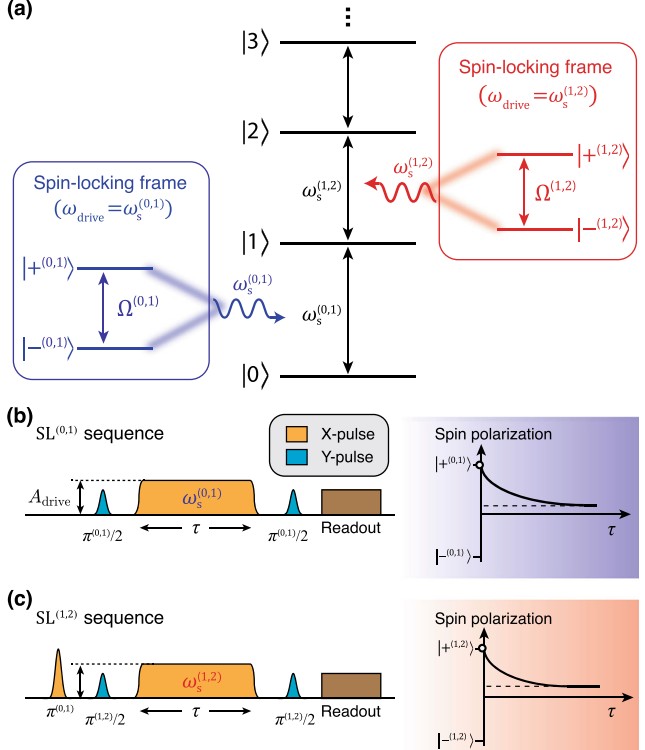

**Fig. 1 Spin-locking noise spectroscopy in a multi-level sensor. a** A transition between the $(j-1)$th and $j$th level of a multi-level system is driven resonantly to form the $j$th spin-locking basis (dressed states) $\{|+^{(j-1,j)}\rangle, |-^{(j-1,j)}\rangle\}$ which are separated by the Rabi frequency $\Omega^{(j-1,j)}$. The two-level system formed by the basis $\{|+^{(j-1,j)}\rangle, |-^{(j-1,j)}\rangle\}$ acts as the $j$th spin-locked noise spectrometer. **b, c** Spin-locking (SL) sequences used to measure the relaxation of spin polarization $|+^{(0,1)}\rangle$ and $|+^{(1,2)}\rangle$ as a function of the spin-locking duration $\tau$, respectively.

spectrometer, respectively. The longitudinal noise in the lab frame (Eq. (1)) for a multi-level system leads to both transverse and longitudinal noise in the spin-locking frame. As a result, the longitudinal noise operator $B^{(j)}(t)$ in the lab frame is transformed into the spin-locking frame as a transverse noise operator $\tilde{B}_\perp^{(j-1,j)}(t)$, which leads to longitudinal relaxation, and the longitudinal noise operator $\tilde{B}_\parallel^{(j-1,j)}(t)$, which leads to transverse relaxation, within the $j$th spin locking subspace. They are given as linear combinations of $B^{(j)}(t)$, arising from the level dressing across multiple levels as follows:

$$\tilde{B}_\perp^{(j-1,j)}(t) = \sum_{k=1}^{d-1} \alpha_{(j-1,j)}^{(k)} B^{(k)}(t), \qquad (4)$$

$$\tilde{B}_\parallel^{(j-1,j)}(t) = \sum_{k=1}^{d-1} \beta_{(j-1,j)}^{(k)} B^{(k)}(t), \qquad (5)$$

where we define the *noise participation ratio* $\alpha_{(j-1,j)}^{(k)}$ $(\beta_{(j-1,j)}^{(k)})$ as a dimensionless factor that quantifies the fraction of the dephasing noise at the $k$th level that is transduced (i.e., projected) to transverse (longitudinal) noise of the $j$th pair of spin-locked states. The noise participation ratios $\alpha_{(j-1,j)}^{(k)}$ and $\beta_{(j-1,j)}^{(k)}$ can be estimated by numerically solving for the dressed states in terms of the bare states $|j\rangle$ (see Supplementary Note 4 for details). Note that the sign of the noise participation ratios can be either positive or negative, leading to the possibility for effective constructive and destructive interference between the noise operators $B^{(k)}(t)$.

There are two noteworthy distinctions between a manifestly two-level system and a multi-level system. First, although the splitting energy $\hbar\Omega^{(j-1,j)}$ between the $j$th pair of dressed states (the $j$th spin-locked states) is *predominantly* determined by the effective driving energy $\hbar(\lambda^{(j-1,j)}A_{\text{drive}})$, they are not universally equivalent. For an ideal two-level system within the rotating wave approximation, the Rabi frequency is indeed proportional to the effective driving field via the standard Rabi formula[16,32,33]. However, this is not generally the case in a multi-level setting due to additional level repulsion from adjacent dressed states[36]. Rather, in the multi-level setting of relevance here, the distinction between $\Omega^{(j-1,j)}$ and $\lambda^{(j-1,j)}A_{\text{drive}}$ must be taken into account to yield an accurate estimation of the noise spectrum.

Second, as a consequence of the multi-level dressing, more than two noise operators $B^{(k)}(t)$ generally contribute to the longitudinal relaxation within a given pair of spin-locked states. In the limit where $\lambda^{(j-1,j)}A_{\text{drive}}$ is small compared to the sensor anharmonicities, Eqs. (4) and (5) reduce to

$$\tilde{B}_\perp^{(j-1,j)}(t) \approx \frac{1}{2}\left[B^{(j-1)}(t) - B^{(j)}(t)\right], \quad \tilde{B}_\parallel^{(j-1,j)}(t) \approx 0, \qquad (6)$$

which conform to the standard spin-locking noise spectroscopy protocol for a two-level sensor[16,32,33]. However, as the effective drive strength $\lambda^{(j-1,j)}A_{\text{drive}}$ increases, the contribution of peripheral bare states—*other* than $|j-1\rangle$ and $|j\rangle$ —to the formation of the spin-locked states $|+^{(j-1,j)}\rangle$ and $|-^{(j-1,j)}\rangle$ increases. As a result, in the large $\lambda^{(j-1,j)}A_{\text{drive}}$ limit, the multi-level dressing transduces the frequency fluctuations of more than two levels to the longitudinal relaxation within the $j$th spin locking frame. Also, this multi-level effect contributes to the emergence of non-zero transverse relaxation $B_\parallel^{(j-1,j)}(t)$, terms which would otherwise be absent within a two-level approximation[16,32,33].

**Noise spectroscopy protocol**. The multi-level noise spectroscopy protocol introduced here consists of measuring the energy decay rate $\Gamma_{1\rho}^{(j-1,j)}$ (i.e., longitudinal relaxation rate) and the polarization $\langle\tilde{\sigma}_z^{(j-1,j)}(\tau)\rangle$ in the $j$th spin-locking frame, and then uses these quantities to extract the spectral density $\tilde{S}_\perp^{(j-1,j)}$ of the noise transverse to the spin-locking quantization axis. This in turn can be related to the longitudinal spectral density $S_\parallel^{(j-1,j)}$ that causes dephasing (i.e., transverse relaxation) in the original, undriven reference frame (the qubit "lab frame"[16]).

We begin by preparing the multi-level sensor in the $j$th spin-locked state $|+^{(j-1,j)}\rangle$ by applying a sequence of resonant $\pi$ pulses $\left[\pi^{(0,1)}, \pi^{(1,2)}, \cdots \pi^{(j-2,j-1)}\right]$, which act to sequentially excite the sensor from the ground state $|0\rangle$ to state $|j-1\rangle$. We then apply a $\pi^{(j-1,j)}/2$ pulse along the $y$-axis of the Bloch sphere, where the north and south poles now correspond to $|j-1\rangle$ and $|j\rangle$, respectively[13]. The pulse acts to rotate the Bloch vector from the south pole to the $x$-axis, thereby placing the multi-level sensor in the $j$th spin-locked state $|+^{(j-1,j)}\rangle = (|j-1\rangle + |j\rangle)/\sqrt{2}$. Subsequently, a spin-locking drive with amplitude $A_{\text{drive}}$ is applied along the $x$-axis (collinear with the Bloch vector) at a frequency resonant with the $|j-1\rangle$-$|j\rangle$ transition and for a duration $\tau$. By adiabatically turning on and off the drive, we keep the state of the sensor within the $j$th spin-locking subspace. Once the drive is off, a second $\pi^{(j-1,j)}/2$ pulse is applied along the $y$-axis in order to map the spin-locking basis $\{|+^{(j-1,j)}\rangle, |-^{(j-1,j)}\rangle\}$ onto the measurement basis $\{|j\rangle, |j-1\rangle\}$, and the qubit is then read out. This procedure is then repeated $N$ times to obtain estimates for the probability of being in states $\{|j\rangle$ and $|j-1\rangle\}$, which

represent the probability of being in states $\{|+^{(j-1,j)}\rangle$ and $|-^{(j-1,j)}\rangle\}$, respectively.

The above protocol is then repeated as a function of $\tau$ in order to measure the longitudinal spin-relaxation decay-function of the $j$th spin-locked spectrometer. For each $\tau$, we define a normalized polarization of the spectrometer,

$$\langle \tilde{\sigma}_z^{(j-1,j)}(\tau)\rangle \equiv \frac{\rho^{(j-1,j-1)}(\tau) - \rho^{(j,j)}(\tau)}{\rho^{(j-1,j-1)}(\tau) + \rho^{(j,j)}(\tau)}, \tag{7}$$

where $\rho^{(j,j)}(\tau)$ denotes the population (the probability) of the $j$th level. From the $\tau$-dependence of $\langle \tilde{\sigma}_z^{(j-1,j)}(\tau)\rangle$, we extract both the relaxation rate $\Gamma_{1\rho}^{(j-1,j)}$ of the spin polarization and the equilibrium polarization $\tilde{\sigma}_z^{(j-1,j)}(\tau)|_{\tau\to\infty}$. The values $\Gamma_{1\rho}^{(j-1,j)}$ and $\tilde{\sigma}_z^{(j-1,j)}(\tau)|_{\tau\to\infty}$ extracted from an experiment performed at a particular Rabi frequency $\Omega^{(j-1,j)}$ are related to the transverse noise PSD $\tilde{S}_\perp^{(j-1,j)}(\omega)$ at angular frequency $\omega = \Omega^{(j-1,j)}$ as follows (see Supplementary Note 5 for details):

$$\Gamma_{1\rho}^{(j-1,j)} = \tilde{S}_\perp^{(j-1,j)}(\omega) + \tilde{S}_\perp^{(j-1,j)}(-\omega), \tag{8}$$

$$\langle \tilde{\sigma}_z^{(j-1,j)}(t)\rangle|_{t\to\infty} = \frac{\tilde{S}_\perp^{(j-1,j)}(\omega) - \tilde{S}_\perp^{(j-1,j)}(-\omega)}{\tilde{S}_\perp^{(j-1,j)}(\omega) + \tilde{S}_\perp^{(j-1,j)}(-\omega)}. \tag{9}$$

Here, the transverse noise spectrum $\tilde{S}_\perp^{(j-1,j)}(\omega)$ is the Fourier transform of the two-time correlation function of the transverse noise operators acting on the spectrometer:

$$\tilde{S}_\perp^{(j-1,j)}(\omega) = \int_{-\infty}^{\infty} d\tau e^{-i\omega\tau}\langle \tilde{B}_\perp^{(j-1,j)}(\tau)\tilde{B}_\perp^{(j-1,j)}(0)\rangle. \tag{10}$$

In the following noise spectroscopy measurements, we will record the spin relaxation for the 1st and 2nd spin-locked noise spectrometers (Fig. 1b, c). Then, the traces are fit to an exponential decay, allowing us to extract $\Gamma_{1\rho}^{(j-1,j)}$ and $\langle \tilde{\sigma}_z^{(j-1,j)}(t)\rangle|_{t\to\infty}$. This is repeated for various drive amplitude $A_{\text{drive}}$ in order to reconstruct $\tilde{S}_\perp^{(j-1,j)}(\omega)$. For simplicity, we will hereafter refer to the spin-locking noise spectroscopy exploiting the $|j-1\rangle-|j\rangle$ transition as SL$^{(j-1,j)}$. To validate the protocol, we will perform the spin relaxation experiments both in the presence and in the absence of engineered noise, and distinguish the contributions of $T_1$ decay and native dephasing noise from the estimation of $\tilde{S}_\perp^{(j-1,j)}(\omega)$ (see Supplementary Note 7 for details).

**Experimental validation.** We use the Xmon[37] variant of the superconducting flux-tunable transmon as a multi-level noise sensor. Our experimental test bed contains three transmon qubits, each of which is dispersively coupled to a coplanar-waveguide cavity for qubit state readout[38,39]. In Fig. 2a, b, the rightmost transmon (blue) operates as a multi-level quantum sensor. The other transmons' modes are far-detuned from the sensor, such that their presence can be neglected (see Supplementary Note 1). In this work, we focus on two environmental noise channels that couple to the transmon sensor. One noise channel is formed by the inductive coupling of the sensor's SQUID loop to the fluctuating magnetic field in the qubit environment (flux noise). In this case, a fluctuating magnetic flux threading the SQUID loop results in the fluctuation of the qubit effective Josephson energy, thereby fluctuating the energy levels of the transmon sensor. The other noise source arises from photon number fluctuations in the readout resonator. In this case, photon-number fluctuations in the readout resonator cause a photon-number-dependent frequency shift of the energy levels of the sensor. Figure 2c shows a reduced measurement schematic.

We generate and apply a known level of engineered flux noise and coherent photon shot noise to the qubit, which we then use as a sensor to validate our protocol (see Supplementary Note 2). We bias the transmon sensor at a flux-sensitive value $\Phi_{\text{ext}} = 0.17\Phi_0$ (dashed line in Fig. 2d). At this operating point, the energy relaxation times $T_1$ for $|0\rangle-|1\rangle$ and $|1\rangle-|2\rangle$ transitions are $\sim 58\mu s$ and $\sim 31\mu s$, respectively. Note that the energy relaxation time for the $|1\rangle-|2\rangle$ transition is approximately half that of the $|0\rangle-|1\rangle$ transition's relaxation time, which is expected for weakly anharmonic systems[40].

To test our protocol, we first demonstrate an accurate spectral reconstruction of engineered flux noise over a range of frequencies – 50 MHz to 300 MHz – that are smaller than, comparable to, and larger in magnitude than the transmon anharmonicity $(\omega_s^{(1,2)} - \omega_s^{(0,1)})/2\pi = -207.3$ MHz. As with the standard spin-locking protocol, the transmon needs to be driven sufficiently strongly to form an energy splitting $\hbar\Omega^{(j-1,j)}$ between a pair of spin-locked states $\{|+^{(j-1,j)}\rangle, |-^{(j-1,j)}\rangle\}$ at the measurement frequency of interest. However, when the splitting energy is comparable with or larger than the anharmonicity, the driven transmon can no longer be approximated as a two-level system, and the multi-level dressing that results must be carefully incorporated into the analysis to accurately reconstruct the PSD.

The first step in our noise spectrosopy demonstration is to measure the Rabi frequencies $\Omega^{(j-1,j)}$ for the $|0\rangle-|1\rangle$ and $|1\rangle-|2\rangle$ transitions as a function of the drive amplitude $A_{\text{drive}}$ (Fig. 3a). To determine $A_{\text{drive}}$, we assume a linear dependence in the weak driving limit where Rabi frequency < 5 MHz. From this linear dependence, we could extrapolate $A_{\text{drive}}$ to the strong driving regime. For both the Rabi and the SL$^{(j-1,j)}$ measurements, the rising and falling edges of the spin-locking drive envelope is Gaussian-shaped $(\propto \exp(-t^2/2\sigma^2))$ with $\sigma = 12$ ns. For a given amplitude, the resulting Rabi frequency for the $|j-1\rangle-|j\rangle$ transition is equivalent to the level splitting $\Omega^{(j-1,j)}$ between the spin-locked states $(|+^{(j-1,j)}\rangle, |-^{(j-1,j)}\rangle)$. However, recall that the measured Rabi frequencies $\Omega^{(j-1,j)}$ begin to deviate from the two-level system approximation $(\Omega^{(j-1,j)} = \lambda^{(j-1,j)}A_{\text{drive}})$ as the drive amplitude is increased. The discrepancy $(\Omega^{(j-1,j)} - \lambda^{(j-1,j)}A_{\text{drive}})$ is due to the multi-level dressing effect, the influence of other levels beyond the two-level approximation. Alternatively, one can also observe such frequency deviations by using pump-probe spectroscopy techniques (see Supplementary Note 3)[35,41]. As such, the frequency shifts $(\Omega^{(j-1,j)} - \lambda^{(j-1,j)}A_{\text{drive}})$ due to this multi-level dressing must be taken into account in order to obtain an accurate estimation of the flux noise spectra at frequencies comparable or larger than the anharmonicity. In our experiments, we found that including up to the 4th excited state [solid curves in Fig. 3a] was sufficient to obtain agreement between our numerical simulations and the experimentally observed frequency shifts.

Similarly, we must consider the noise participation of the peripheral bare states introduced through the multi-level dressing effect in order to obtain an accurate spectral esimation at high frequencies. To build intuition, we begin considering the low-frequency (small $A_{\text{drive}}$) limit, where the longitudinal relaxation for SL$^{(j-1,j)}$ is determined solely by dephasing noise that acts on $|j-1\rangle$ and $|j\rangle$ (Eq. (6)). Then, in the large $A_{\text{drive}}$ limit, the flux noise acting on the peripheral levels also contributes to the longitudinal spin relaxation. Thus, we must use the noise participation ratios $\alpha_{(j-1,j)}^{(k)}$ for each energy level $k$, including the original two levels and the peripheral levels:

$$S_\Phi(\omega) = \tilde{S}_{\Phi,\perp}^{(j-1,j)}(\omega) \times \left(\sum_k \alpha_{(j-1,j)}^{(k)}\frac{\partial\omega_s^{(k)}}{\partial\Phi_{\text{ext}}}\right)^{-1}, \tag{11}$$

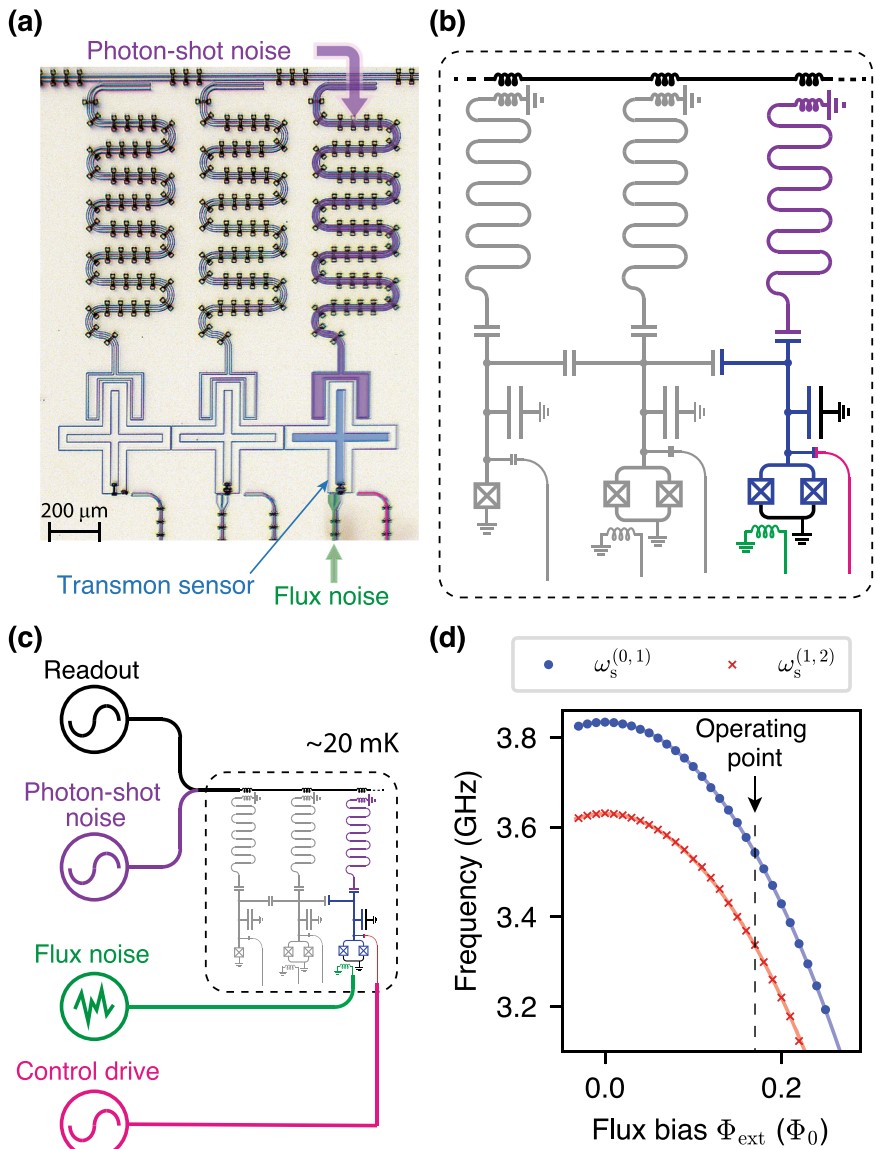

**Fig. 2 Device layout and simplifed experimental setup. a** Optical micrograph (false color) of the superconducting circuit comprising a flux-tunable transmon sensor (blue) to measure flux noise and photon shot noise applied via independent channels (green and purple, respectively). The transmon is controlled via a capacitively coupled drive line (magenta). **b** Circuit schematic. The additional transmon qubits (gray) are far detuned from the frequency of the transmon sensor and can be neglected in this experiment. **c** Simplified measurement schematic. Known, engineered flux noise and photon-shot noise is applied to the qubit. The control and readout lines are used to perform noise spectroscopy protocol and measure the results. **d** $|0\rangle$–$|1\rangle$ transition frequency (blue circles) and $|1\rangle$–$|2\rangle$ transition frequency (red crosses) of the transmon sensor as a function of the external flux bias $\Phi_{\text{ext}}$. Solid lines correspond to simulations based on the circuit parameters (see Supplementary Note 1). The transmon sensor operates at a flux-sensitive point, $\Phi_{\text{ext}} = 0.17\Phi_0$, See dashed black line.

where $S_\Phi(\omega)$ denotes the power spectral density of the engineered flux noise at frequency $\omega/2\pi$, and $\partial\omega_s^{(k)}/\partial\Phi_{\text{ext}}$ denotes the flux noise sensitivity of the $k$th level of the sensor. For our experiment, the values of $\alpha_{(j-1,j)}^{(k)}$ for $\text{SL}^{(0,1)}$ and $\text{SL}^{(1,2)}$ are numerically estimated and shown in Fig. 3b, c, respectively. We also numerically estimate $\partial\omega_s^{(k)}/\partial\Phi_{\text{ext}}$ for $k \in \{1, \cdots, 4\}$ by solving the circuit Hamiltonian of the transmon sensor (see Supplementary Note 1).

We now reconstruct the spectrum of engineered Lorenzian-distributed flux noise centered at 200 MHz, a frequency comparable to the sensor anharmonicity (see in Fig. 3d). For the sake of comparison, we first plot PSD estimates based on a two-level approximation (hollow circles). The frequencies of these PSD estimates are shifted by $\lambda^{(j-1,j)}A_{\text{drive}} - \Omega^{(j-1,j)}$ from the ideal flux

noise spectra (gray shading). We would also conclude (erroneously) that the extracted flux noise PSD amplitude increases as the frequency increases when estimated using the two-level approximation. In order to estimate the flux noise PSD accurately, the two corrections described above must be applied to the PSD estimates to account for the multi-level dressing effects: Step 1 – a frequency shift; and Step 2 – an amplitude adjustment. Upon applying these corrections, we successfully reconstruct the PSD estimates for the 200 MHz engineered flux noise for both $\text{SL}^{(0,1)}$ and $\text{SL}^{(1,2)}$ (markers lie on gray region, Fig. 3d).

Using this approach, we benchmark the performance of $\text{SL}^{(0,1)}$ and $\text{SL}^{(1,2)}$ for a set of the Lorentzian-shaped engineered flux noise spectra which are centered at $f_0 = 50, 100, 150, 200, 250,$ and 300 MHz. Figure 3e, f compare the ideal noise spectra (gray

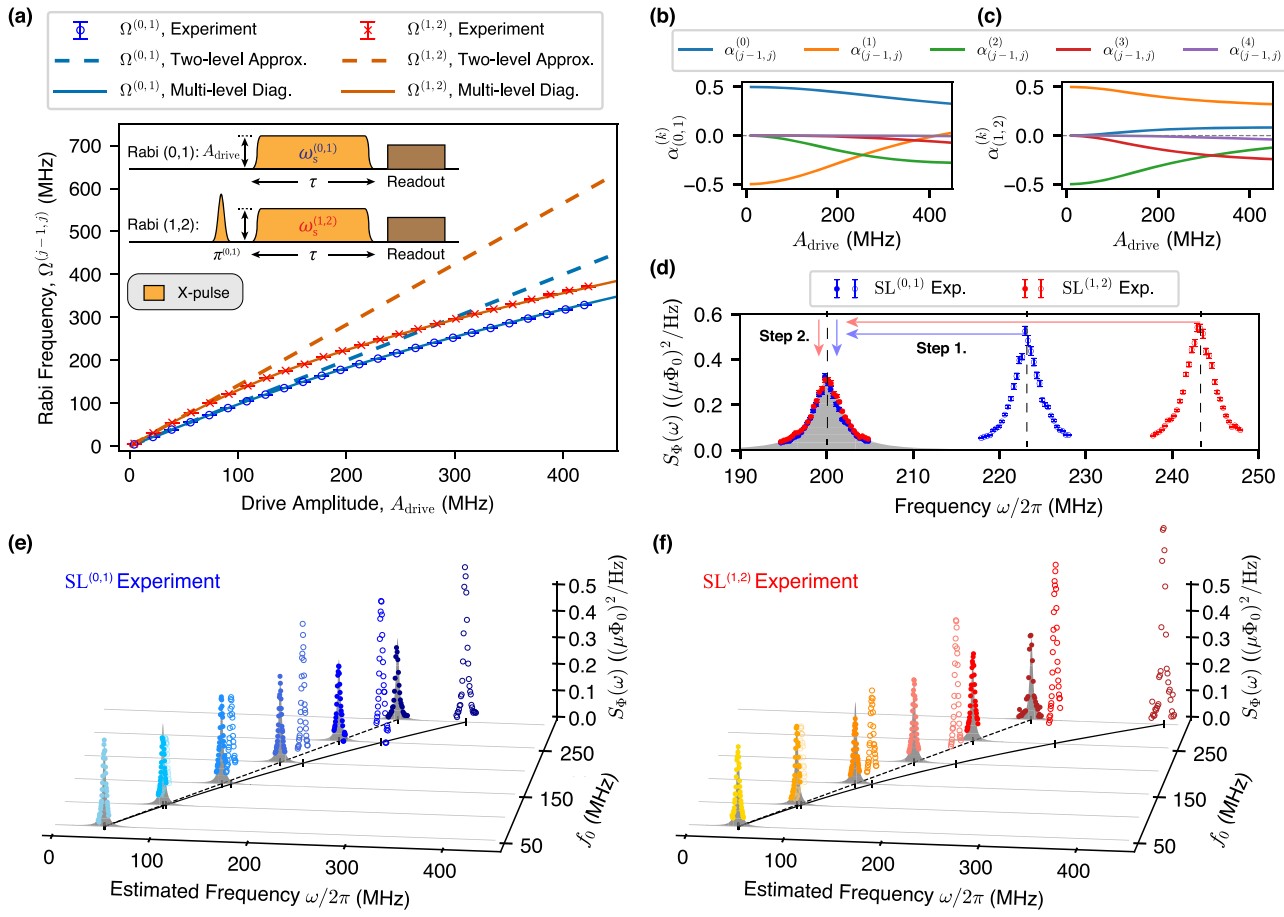

**Fig. 3 Accurate spectral estimation of high-frequency noise. a** Rabi frequencies $\Omega^{(j-1,j)}$ for $|0\rangle$–$|1\rangle$ (blue) and $|1\rangle$–$|2\rangle$ (red) transitions as a function of the drive amplitude $A_{\text{drive}}$. Inset: pulse sequences used to measure the Rabi frequencies. **b**–**c** Numerical calculations of noise participation ratios $\alpha_{(j-1,j)}^{(k)}$ as a function of drive amplitude $A_{\text{drive}}$. **d** Spectral estimation of the engineered flux noise (Lorentzian, centered at 200 MHz) via $SL^{(0,1)}$ (blue) and $SL^{(1,2)}$ (red) experiments. Two corrections are applied to the estimates under the two-level approximation (hollow circles), and consist of shifting the frequency (step 1) and adjusting the magnitude (step 2) due to the multi-level dressing. Note that the corrected flux noise spectra for $SL^{(0,1)}$ (blue circles) and $SL^{(1,2)}$ (red circles) are in good agreement with the ideal flux noise PSD (gray filled). Error bars represent ±1 standard deviations. **e**, **f** Benchmarking the spectral estimation of engineered flux noise ranging from $f_0 = 50$ MHz to $f_0 = 300$ MHz, where $f_0$ corresponds to the center frequency of engineered noise spectra. The different color shades of the PSD estimates correspond to engineered flux noise with different center frequencies. The agreement between the corrected experimental estimates (circles) and the ideal flux noise PSDs (gray filled) indicates that our protocol overcomes the spectral limit imposed by the sensor anharmonicity.

shading) with the corrected flux noise PSD estimates (circles sitting on the envelope of the gray regions and following a dashed line) measured by $SL^{(0,1)}$ (blue shades) and $SL^{(1,2)}$ (red shades), respectively, and with the uncorrected estimates ("x" shapes following a solid line) that deviate in both the inferred frequency and power. The different colors correspond to the different engineered flux noise spectra. The agreement between corrected PSD estimates and the engineered noise PSDs clearly substantiates the idea that our protocol overcomes the anharmonicity limit of the noise sampling frequency by taking the multi-level dressing effect into account.

We now move on to distinguishing the noise contributions from both engineered flux and photon-shot noise by measuring $SL^{(0,1)}$ and $SL^{(1,2)}$. Importantly, both noise sources induce frequency fluctuations of the $|0\rangle$–$|1\rangle$ and $|1\rangle$–$|2\rangle$ transitions, but with a different and distinguishing relative noise power ($\tilde{S}_\perp^{(1,2)}(\omega)/\tilde{S}_\perp^{(0,1)}(\omega)$).

In the case of flux noise, since the degree of transmon anharmonicity is independent of the external magnetic flux threading the transmon loop $\Phi_{\text{ext}}$[26], the flux-noise-induced frequency fluctuations of the $|0\rangle$–$|1\rangle$ and $|1\rangle$–$|2\rangle$ transitions are

equal: $\partial\omega_s^{(0,1)}/\partial\Phi_{\text{ext}} = \partial\omega_s^{(1,2)}/\partial\Phi_{\text{ext}}$. Therefore, for low-frequency flux noise that causes dephasing, the relative noise power spectra of $SL^{(1,2)}$ to $SL^{(0,1)}$ is given as:

$$\frac{\tilde{S}_{\Phi,\perp}^{(1,2)}(\omega)}{\tilde{S}_{\Phi,\perp}^{(0,1)}(\omega)} = 1, \tag{12}$$

where we have introduced the subscript $\Phi$ to indicate flux noise due to $\Phi_{\text{ext}}$.

In contrast, photon-shot noise induces frequency fluctuations for each level transition that scale with the corresponding effective dispersive strength $\chi^{(j-1,j)}$[32]. The photon-number-dependent frequency shift due to photon shot noise affecting the $|j-1\rangle$–$|j\rangle$ transition is given as $\delta\omega_s^{(j-1,j)} = 2\chi^{(j-1,j)}\bar{n}$, where $\bar{n}$ is the average residual photon number in the resonator. Hence, the relative noise power spectra of $SL^{(1,2)}$ to $SL^{(0,1)}$ for photon shot noise is:

$$\frac{\tilde{S}_{\bar{n},\perp}^{(1,2)}(\omega)}{\tilde{S}_{\bar{n},\perp}^{(0,1)}(\omega)} = \left(\frac{\chi^{(1,2)}}{\chi^{(0,1)}}\right)^2, \tag{13}$$

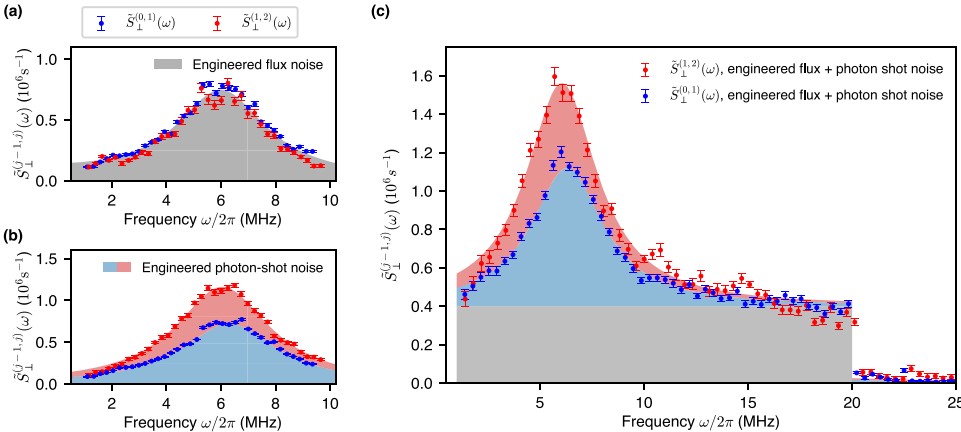

**Fig. 4 Distinguishing the noise contributions from flux and photon shot noise. a** Transverse flux noise PSDs $\tilde{S}_{\Phi,\perp}^{(0,1)}(\omega)$ (blue) and $\tilde{S}_{\Phi,\perp}^{(1,2)}(\omega)$ (red) measured by performing $SL^{(0,1)}$, $SL^{(1,2)}$ for engineered Lorenztian flux noise (gray filled) centered at 6 MHz. **b** Transverse photon shot noise PSDs $\tilde{S}_{\bar{n},\perp}^{(0,1)}(\omega)$ (blue) and $\tilde{S}_{\bar{n},\perp}^{(1,2)}(\omega)$ (red) for engineered photon shot noise with detuning $\Delta/2\pi = 6.05$ MHz from the readout resonator. **c** Total transverse noise PSDs $\tilde{S}_{\perp}^{(0,1)}(\omega)$ (blue) and $\tilde{S}_{\perp}^{(1,2)}(\omega)$ (red) for a mixture of engineered flux noise (gray-shaded box-car, 1 MHz to 20 MHz), and coherent photon shot noise with detuning $\Delta/2\pi = 6.05$ MHz (blue- and red-shaded Lorentzians). Measuring the twofold noise spectra $\tilde{S}_{\perp}^{(0,1)}(\omega)$ and $\tilde{S}_{\perp}^{(1,2)}(\omega)$ distinguishes the flux noise and photon shot noise contributions. Error bars represent ±1 standard deviations.

where we have introduced the subscript $\bar{n}$ to indicate photon shot noise. This finding highlights the usefulness of measuring multiple noise spectra in order to deconvolve environmental noise processes. We shall presume that these two independent sources of engineered noise—flux noise and photon shot noise—are the only two sources of transverse noise impacting the spin-locked spectrometers, and so we may define the total transverse noise power spectrum as $\tilde{S}_{\perp}^{(j-1,j)}(\omega) = \tilde{S}_{\Phi,\perp}^{(j-1,j)}(\omega) + \tilde{S}_{\bar{n},\perp}^{(j-1,j)}(\omega)$.

We now demonstrate the identification and characterization of two independent noise sources by measuring the twofold noise spectra $\tilde{S}_{\perp}^{(0,1)}(\omega)$ and $\tilde{S}_{\perp}^{(1,2)}(\omega)$. To begin, in Fig. 4a, we present the experimentally extracted spectra $\tilde{S}_{\Phi,\perp}^{(0,1)}(\omega)$ (blue circles) and $\tilde{S}_{\Phi,\perp}^{(1,2)}(\omega)$ (red circles) for solely Lorentzian-shaped engineered flux noise centered at 6 MHz. The measured $\tilde{S}_{\Phi,\perp}^{(0,1)}(\omega)$ and $\tilde{S}_{\Phi,\perp}^{(1,2)}(\omega)$ are essentially identical, as one expects for transmon flux noise and consistent with Eq. (12). Similarly, we also present the extracted $\tilde{S}_{\bar{n},\perp}^{(0,1)}(\omega)$ and $\tilde{S}_{\bar{n},\perp}^{(1,2)}(\omega)$ for solely engineered coherent photon-shot noise. The results are Lorentzian-shaped spectra centered at the frequency detuning $\Delta/2\pi \equiv (\omega_r - \omega_n)/2\pi = 6.05$ MHz between the readout resonator resonance frequency ($\omega_r/2\pi$) and the applied coherent tone frequency ($\omega_n/2\pi$) used to generate the shot noise (Fig. 4b)[32]. Contrary to the flux noise case, here the measured $\tilde{S}_{\bar{n},\perp}^{(0,1)}(\omega)$ and $\tilde{S}_{\bar{n},\perp}^{(1,2)}(\omega)$ have different magnitudes, with a measured ratio of $\tilde{S}_{\bar{n},\perp}^{(1,2)}(\omega)/\tilde{S}_{\bar{n},\perp}^{(0,1)}(\omega) \approx 1.61$ (see Eq. (13)), due to the differing values of $\chi^{(0,1)}$ and $\chi^{(1,2)}$. Next, we demonstrate the noise spectroscopy of a mixture of two engineered noise and identify their individual noise contributions. We inject flux noise ranging from 1 MHz to 20 MHz with a "box-car" envelope and, simultaneously, coherent photon shot noise from a coherent tone with a frequency that is detuned by $\Delta/2\pi = 6.05$ MHz from the readout-resonator resonance frequency. Figure 4c presents the experimental data for the total transverse noise power spectra $\tilde{S}_{\perp}^{(0,1)}(\omega)$ (blue) and $\tilde{S}_{\perp}^{(1,2)}(\omega)$ (red), which include both flux noise and photon shot noise contributions. The known, engineered noise spectra for each transition is indicated by the gray box car (flux noise) and by the blue and red Lorentzians (shot noise) for

the 0-1 and 1-2 transitions, respectively. Over the frequency domain where both flux and photon-shot noise are significant, ($3 \text{ MHz} \leq \omega/2\pi \leq 9 \text{ MHz}$), a distinction between $\tilde{S}_{\perp}^{(0,1)}(\omega)$ and $\tilde{S}_{\perp}^{(1,2)}(\omega)$ is clearly observed, and the measured total noise spectral density is the sum of the flux noise and photon shot noise contributions, consistent with our assumption that these two noise sources are independent. In contrast, over the frequency domain where flux noise dominates ($15 \text{ MHz} \leq \omega/2\pi \leq 20 \text{ MHz}$), $\tilde{S}_{\perp}^{(0,1)}(\omega)$ and $\tilde{S}_{\perp}^{(1,2)}(\omega)$ are similar in magnitude and predominantly match the gray region. Lastly, at 20 MHz, above which no external noise was applied, the data exhibit a discrete jump down to the sensitivity limit of the experiment (See Supplementary Note 6 for discussion on the sensitivity limit due to $T_1$ of the sensor). This result indicates that we can distinguish the noise contributions from flux and photon shot noise by measuring the twofold noise spectra $\tilde{S}_{\perp}^{(0,1)}(\omega)$ and $\tilde{S}_{\perp}^{(1,2)}(\omega)$. More generally, the independent extraction of unknown flux noise and photon shot noise would be performed by measuring $\tilde{S}_{\perp}^{(j-1,j)}(\omega)$ for a sufficient number of transitions $j-1, j$ and frequency ranges in order to back out the individual contributions (within certain and appropriate assumptions about the origin and type of noise).

## Discussion

In summary, we introduced and experimentally validated a noise spectroscopy protocol that utilizes multiple transitions of a qubit as a quantum sensor of its noise environment. By moving beyond the conventional two-level approximation, our approach overcomes the anharmonicity frequency limit of previous spin-locking approaches. We further show that measuring the noise spectra for multiple transitions enables one to distinguish certain noise sources, such as flux noise and photon shot noise, by leveraging the differing impact of those noise sources on the different transitions. As an example, we measured the twofold power spectra of dephasing noise acting on the $|0\rangle$–$|1\rangle$ and $|1\rangle$–$|2\rangle$ transitions of a transmon, and showed that our protocol can distinguish between externally applied, known, engineered noise contributions from flux noise and photon shot noise. We anticipate that applying this protocol to even higher level

transitions ($j > 2$) of a superconducting qubit sensor will enable one to distinguish other dephasing noise sources, such as charge noise[42].

Although we mainly focus on the spin-locking based multi-level QNS throughout this work, extending the dynamic decoupling (D.D.) based noise spectroscopy protocols[6,9,24] to multi-level systems would also yield improved QNS performance (see Supplementary Note 8 for a discussion of why we focus on the spin-locking based approaches rather than the D.D. based approaches throughout this work). Notably, the idea of discriminating noise sources by employing multiple level transitions as distinct spectrometers is immediately applicable to dynamic decoupling based approaches. In view of recent advances in optimal band-limited control[43], we expect the implementation of dynamic decoupling based QNS using multi-level sensors will augment knowledge about noise sources in a manner similar to the spin-locking approach described here.

In this paper, we demonstrated our protocol by measuring engineered noise in the flux-tunable transmon sensor. We chose the operating point (flux bias) and operating frequency range (measured spectral range) of the sensor, such that it is dominantly affected by the engineered noise. However, the technique discussed here can be also applied to measure intrinsic noise of transmons such as $1/f$ flux noise[3,44,45]. Notably, by biasing the sensor at more flux-sensitive point, the sensitivity to flux noise can be further increased in order to detect intrinsic flux noise.

While we employ a flux-tunable transmon as a multi-level noise sensor, our methodology is portable to other anharmonic multi-level systems, such as the C-shunt flux qubit[30] and the fluxonium[46,47]. Since the sensitivity of the qubit energies to various noise sources differ by qubit design, employing other superconducting qubits as multi-level noise sensors will enable us to explore various noise sources. We also envision the spin-locking QNS protocols - whether in a TLS approximation or a multi-level system - being used for other qubit modalities, such as quantum dot qubits or trapped ion qubits, as sensors of their local environments, such as their substrates or surface traps.

As detailed in Supplementary Note 6, the $T_1$ of the qubit can limit its noise sensitivity. However, as $T_1$ is improved through a combination of qubit design[47] and advanced materials[48], the sensitivity and utility of our approach also improves. Using diagnostic techniques such as the QNS protocol developed here to identify and characterize noise sources is an important step towards mitigating and eliminating them.

## Data availability
The data that support the findings of this study may be made available from the corresponding authors upon request and with the permission of the US Government sponsors who funded the work. Source data are provided with this paper.

## Code availability
The code used for the analyses may be made available from the corresponding authors upon request and with the permission of the US Government sponsors who funded the work.

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

## Acknowledgements

It is a pleasure to thank F. Beaudoin, L. M. Norris, and L. Viola for insightful discussions, and M. Pulido for generous assistance. This research was funded by the U.S. Army Research Office grant No. W911NF-14-1-0682; and by the Department of Defense via MIT Lincoln Laboratory under Air Force Contract No. FA8721-05-C-0002. Y.S. acknowledges support from the Korea Foundation for Advanced Studies. The views and conclusions contained herein are those of the authors and should not be interpreted as necessarily representing the official policies or endorsements, either expressed or implied, of the U.S. Government.

## Author contributions

Y.S., A.V., J.B., and W.D.O conceived the project. Y.S., A.V., and S.G. performed the experiment and F.Y. and W.D.O. provided feedback. Y.S. developed the theoretical framework and carried out the numerical simulation with constructive feedback from A.V., J.B., F.Y., and W.D.O. A.J.M., D.K.K., and J.L.Y. fabricated the device. J.B., J.W., M.K., R.W., A.B., and M.E.S. provided experimental assistance. T.P.O., S.G., and W.D.O. supervised the project. All authors contributed to the discussion of the results and the manuscript.

## Competing interests

Y.S., J.B., A.V., S.G., W.D.O., and Massachusetts Institute of Technology have filed a provisional US patent application related to multi-level quantum noise spectroscopy protocols. All other authors have no competing interests.
