## [Peer Review File · Nature Communications]

REVIEWER COMMENTS

Reviewer #1 (Remarks to the Author):

Mitigating the individual noise sources for each quantum computing platform is critical for advancing these technologies. In this manuscript, the authors extend a mainstay for two-level system noise characterization (spin locking) to weakly anharmonic d-level atoms, such as the ubiquitous transmon type of superconducting circuit. This work builds on the group's previous experiments involving noise analysis with spin locking [Yan et al Nat. Commun. (2013)] and decoherence in multi-level excitations of transmons [Peterer et al PRL (2015)]. Their current results tie these two concepts together into a consistent framework. In achieving that, the authors are able to estimate the power spectral density (PSD) at frequencies approximately equal to the anharmonicity of a three-level system, a previously inaccessible high-frequency regime for PSD estimation. The authors also introduce a clever application of this technique to separate two types of noise common in superconducting devices. This example nicely illustrates the interplay between novel noise-spectroscopy and possible new insights. The manuscript is likely to be of wide interest to quantum information scientists, and I recommend Nature Communications or another suitable journal accept it for publication. The following suggestions may also improve the manuscript.

Consider explaining how this treatment works (and why it is needed) in more technical detail in the introduction of the manuscript. I believe the point is well illustrated in Figure 3d and starting on page six "We now reconstruct the spectrum...". I would encourage the authors to explain that by treating dressing of higher-levels (up to $d=4$), the method corrects a diverging systematic error, skewing noise to higher frequencies. The current, high-level description in the manuscript leaves too much up to the imagination ahead of the Noise Spectroscopy Protocol section. For instance, that these drives will be adiabatically ramped to remain in the j -th spin locking basis is important context for the maths contained earlier.

Because the manuscript considers only single-photon transitions (between levels $j-1$ and j) - it would substantially clarify the notation to label terms with single indexing, instead of the cumbersome $(j-1, j)$ super/sub-scripts throughout. This would make the manuscript more consistent with conventions in e.g. atomic systems (where Ω_j typically addresses the $(j-1, j)$ transition), and thus increase the breadth of its appeal. The authors already make this leap by defining the " j -th spin locking basis." The potential for confusion in the authors' notation is already present in Eq. 1 which introduces ω^j , only to soon be replaced with a new $\omega^{\{j-1,1\}}$. Unfortunately, for the case of $j=1$, the abuse of notation requires the existence of an ω_0 to define $\omega_{0,1}$, which is not a Hamiltonian term (since the summation begins at $j=1$). Later, the anharmonicity is defined as $\omega_{1,2} - \omega_{0,1}$ which is correct, but since the anharmonicity is a bare Hamiltonian term, the current indexing strategy would have suggested $\omega_2 - \omega_1$.

This treatment is focused on pure dephasing noise. It would be helpful to speculate on what the consequences of energy relaxation would be for this protocol, or at least justify why this assumption is valid for the experiment.

This work considers noise on two of the three driving mechanisms in the circuit (flux, readout). Speculating on what, if anything, could be said about shot noise on the "Control line" would be useful in mastering this type of QNS for system-level analysis.

On page 1, during the listing of qubit platforms that have tested QNS protocols, consider adding two canonical references [NMR: Vandersypen and Chuang RMP (2005); SC Qubits: Ithier et al PRB (2005); and another platform Rydberg atoms [Carter and Martin PRA (2013)] which uses spin locking.

On page 3, in the sentence "As a result, in the large..." possible typo if Ω_{drive} should be A_{drive} , as Ω_{drive} is undefined.

On page 4, Fig. 2 schematic has a "Flux noise" source but not a DC operating point controller.

Reviewer #2 (Remarks to the Author):

The Authors present an experimental and theoretical study of an application of spin-locking protocol of noise spectroscopy to a transmon qubit. This qubit differs from many others due to its weak anharmonicity and the fact that the qubit is based on two out of many energy levels - but one can easily drive the state out of the "qubit subspace" by applying a driving field of a slightly different frequency than the one used to drive the Rabi oscillations within this subspace.

The spin-locking protocol of noise spectroscopy was previously shown in Refs [13] and [29] by some of the Authors of the manuscript. The novel factor in this work is taking into account the multi-level energy spectrum of the transmon device. Its presence is both a nuisance (as one has to abandon the two-level system approximation, and everything gets complicated) with which the Authors deal in a transparent way, but, as the Authors show, it also brings benefits, as one can consider spin-locking within various two-level subspaces, and draw interesting conclusions from comparison of results of noise spectroscopy for distinct subspaces.

The paper describes work that pushes the noise spectroscopy with transmons (and related qubits) into an interesting direction, but I am not fully convinced if the obtained results are going to significantly influence the field, or if they are a slightly involved extension of [13] and [29], in which more realistic structure of the system is taken into account - this is mostly related to point 3 below.

1) In the abstract we can read that the discussed method "extends the spectral range" of noise spectroscopy - but the limitations on this range specific to the more widely used dynamical decoupling noise spectroscopy and on the other hand to the spin-locking method are not discussed in the paper, and no citation is given to another work in which this is investigated. This is a clear omission: the reader is not given a justification why one should rather use spin locking instead of e.g. dynamical decoupling to characterize dephasing noise (in the original Hamiltonian) for transmons.

2) The issue of what terms couple longitudinally/transversely in the original Hamiltonian / in the Hamiltonian using the basis of dressed states, and their relation to longitudinal/transverse relaxation could be perceived as confusing by the reader. First, while in the appendix it is clearly stated that "longitudinal noise in the lab frame for a multi-level system leads to both transverse and longitudinal noise in the spin-locking frame", such a simple statement seems to be missing from the main text, where it would be quite useful. Also, while it is known that longitudinally coupled noise causes transverse relaxation (and transversely coupled noise causes longitudinal relaxation), having both the terminology such as the one below Eq (1): "the $B(j)(t)$ correspond to the time-dependent noise operators that longitudinally couple to the j -th level of the sensor", and then "the emergence of non-zero transverse relaxation $B_{\{j-1;j\}k}(t)$, terms" is confusing. It would be better to call $B_{\{j-1;j\}k}(t)$ longitudinal (in the spin-locked frame) noise that causes transverse relaxation.

3) Why only the influence of engineered noise was investigated with the proposed method? Isn't there anything to say about the intrinsic noise? After the spectrometry protocol is validated for a known injected noise spectrum, it could be used for additional characterization of the intrinsic noise affecting the qubit. In the appendix it is discussed how the influence of that noise is subtracted from results to reconstruct the engineered noise. But why not characterize the intrinsic noise with the discussed method? At least a comment would be welcome, and showing some result would make the paper much stronger.

This question is related to a more general issue: the transmon was designed in order to limit the amount of dephasing that the qubit experiences, with the price paid by decreasing the anharmonicity of the system spectrum. Now, in order to sense the longitudinal (in the original frame) noise, one has to do spin locking and consider the multi-level structure of the system. But is the dephasing noise affecting the qubit strong enough/interesting enough to go through all this effort to characterize it?

Reviewer #3 (Remarks to the Author):

Multi-level quantum noise spectroscopy describes a method of detecting and characterizing noise using a multi-level quantum sensor, using a superconducting flux qubit for the experimental demonstration. This is an extension of previous work by Oliver et. al., primarily reference [13], which used spin locking experiments to measure noise in the rotating frame. This paper proposes a method for using multi-level systems (such as a weakly anharmonic flux qubit) for noise spectroscopy. This requires a complete analysis that accounts for the higher levels on the rotating frame of any two driven levels. It also allows the experimentalist to distinguish between different types of noise by probing higher levels.

The essential experiment is to first bring the system to the starting state by a series of pi-pulses and then perform a Ramsey experiment around the application of a spin locking field. Varying the length of the spin locking drive gives the polarization as a function of time and $T_1\rho$. These quantities are enough to calculate the power spectral density for the pair of levels that correspond to the spin locking drive.

The authors demonstrate this method on a flux qubit by measuring the power spectral densities of the 0-1 and 1-2 subspaces. They use engineering flux noise and photon shot noise to validate the method and show that they can distinguish between the two types of noise.

Overall, this work is well presented and clearly demonstrates a new method of noise spectroscopy. The careful theoretical analysis in the supplementary materials is the strength of the paper. The analysis of the effect of higher levels on the rotating frame Hamiltonian is technically sound and the additional description of pump-probe is a useful resource.

I recommend the paper for publication with a few comments:

One concern is that the approach is tailored to flux qubits and the particular kind of noise that may be present in flux qubits (photon shot noise, flux noise). I would suggest adding more commentary on how this methodology could be used in other systems (the authors do mention charge noise using additional higher levels). Additionally, the system used was a qubit that was fairly isolated from other coupled qubits. How would stronger coupling impact the analysis?

The results with engineered noise are convincing, but it would be worth adding some discussion on the sensitivity of this method.

It's not clear in the pump-probe experiment (Fig. S5) and the Rabi experiment (Fig 3(a)), if A_{drive} is an extrapolation from the low frequency measurements assuming a two-level system or if there a fit parameter to the multi-level model to extract A_{drive} . Since the deviation of the Rabi frequency from A_{drive} is important for this work, I think it is worth being explicit about how A_{drive} is determined.

In Fig. 3(d) it is difficult to tell the marker shapes apart, perhaps another shape or different color would make this plot clearer.

Re: NCOMMS-19-08909-T

In the following, **blue** indicates referee text, **black** indicates our response to the referee. All the changes we implemented in the main text and supplement are clearly marked in **red**.

Response to Reviewer #1

Mitigating the individual noise sources for each quantum computing platform is critical for advancing these technologies. In this manuscript, the authors extend a mainstay for two-level system noise characterization (spin locking) to weakly anharmonic d-level atoms, such as the ubiquitous transmon type of superconducting circuit. This work builds on the group's previous experiments involving noise analysis with spin locking [Yan et al Nat. Commun. (2013)] and decoherence in multi-level excitations of transmons [Peterer et al PRL (2015)]. Their current results tie these two concepts together into a consistent framework. In achieving that, the authors are able to estimate the power spectral density (PSD) at frequencies approximately equal to the anharmonicity of a three-level system, a previously inaccessible high-frequency regime for PSD estimation. The authors also introduce a clever application of this technique to separate two types of noise common in superconducting devices. This example nicely illustrates the interplay between novel noise-spectroscopy and possible new insights. The manuscript is likely to be of wide interest to quantum information scientists, and I recommend Nature Communications or another suitable journal accept it for publication. The following suggestions may also improve the manuscript.

We are pleased with the positive assessment and recommendation of the referee, in particular his/her appreciation of the key importance of our achievement toward fully exploiting the potential of quantum technologies. We have carefully considered all of the comments and recommendations and implemented modifications in the revised version of both the manuscript and supplement. Specifically, we provide point-by-point responses in what follows.

1) Consider explaining how this treatment works (and why it is needed) in more technical detail in the introduction of the manuscript. I believe the point is well illustrated in Figure 3d and starting on page six "We now reconstruct the spectrum...". I would encourage the authors to explain that by treating dressing of higher-levels (up to $d=4$), the method corrects a diverging systematic error, skewing noise to higher frequencies. The current, high-level description in the manuscript leaves too much up to the imagination ahead of the Noise Spectroscopy Protocol section. For instance, that these drives will be adiabatically ramped to remain in the j -th spin locking basis is important context for the maths contained earlier.

We thank the referee for this suggestion. We agree that primary problem we are addressing and the conceptual roadmap for the paper should be introduced earlier. In response to this comment, we have added the following sentences in the introduction.

"As with many contemporary QNS protocols, the spin-locking approach presumes a two-level-system approximation. While this approximation holds at low frequencies, its validity breaks down as one attempts to perform noise spectroscopy at frequencies approaching and exceeding qubit anharmonicity (e.g., around 200-300 MHz is conventional superconducting transmon qubits) due to the impact of additional energy levels, leading to systematic errors in the extracted noise spectrum."

“In this work, we develop a multi-level spin-locking QNS protocol and experimentally validate it using a flux-tunable transmon qubit and accounting for five energy levels.”

2) Because the manuscript considers only single-photon transitions (between levels $j-1$ and j) - it would substantially clarify the notation to label terms with single indexing, instead of the cumbersome $(j-1,j)$ super/sub-scripts throughout. This would make the manuscript more consistent with conventions in e.g. atomic systems (where Ω_j typically addresses the $(j-1,j)$ transition), and thus increase the breadth of its appeal. The authors already make this leap by defining the “ j -th spin locking basis.” The potential for confusion in the authors’ notation is already present in Eq. 1 which introduces $\omega^{\{j\}}$, only to soon be replaced with a new $\omega^{\{j-1,1\}}$. Unfortunately, for the case of $j=1$, the abuse of notation requires the existence of an ω_0 to define $\omega_{\{0,1\}}$, which is not a Hamiltonian term (since the summation begins at $j=1$). Later, the anharmonicity is defined as $\omega_{\{1,2\}} - \omega_{\{0,1\}}$ which is correct, but since the anharmonicity is a Hamiltonian term, the current indexing strategy would have suggested $\omega_{\{2\}} - \omega_{\{1\}}$.

We thank the referee for this comment on our notation. We agonized over whether to use a simplified notation – as the referee suggests – or the current notation when we initially wrote the paper. We agree that the superscript $(j-1,j)$ is somewhat cumbersome. However, the manuscript does not only consider adjacent level transitions, but also energies referenced to the ground state. Therefore, we elected to use single indices for levels referenced to the ground state, and double indices to represent adjacent transitions. For example, $\omega^{(j)}$ and $B^{(j)}$ in Eqs. (1) – (5) do not refer to $|j-1\rangle - |j\rangle$ transitions, but rather reference level j to the ground state.

To address the specific case indicated by the referee, we introduced $\omega^{(j)}$ in Eq. (1) to reference the j -th eigenenergy of a multi-level system to the ground state energy (taking $\omega^0=0$). Similarly, in Eqs. (3) – (5), we introduced $B^{(j)}$ to denote the energy fluctuations of the energy level $|j\rangle$ referenced to the ground state $|0\rangle$. These terms do not reference $|j-1\rangle - |j\rangle$ transitions. Therefore, these single-indexed terms are not interchangeable with the present “ $|j-1\rangle - |j\rangle$ terms”. Based on the referee suggestion, we considered using a notation $(0,j)$ to make the notation consistently double-indexed throughout. However, in the interest of simplicity, we elected to stay with just (j) when referencing to the ground state.

3) This treatment is focused on pure dephasing noise. It would be helpful to speculate on what the consequences of energy relaxation would be for this protocol, or at least justify why this assumption is valid for the experiment.

This is indeed an important point, on which we had been insufficiently explicit. In response to the referee’s comment, we now have added “**Supplementary Material 6 (Contributions of T1 Decay to the Spin Relaxation $\Gamma_{1\rho}$)**” to discuss the consequences of energy relaxation in detail. In there, we derive the effective longitudinal spin relaxation ($\Gamma_{1\rho,eff}$) as a consequence of the energy relaxation (T1) of a multi-level qubit. In contrast with previous studies on the spin-locking spectroscopy [1, 2], we take into account multi-level effects.

It is justified to focus on pure dephasing noise in our experiments, since

1. The injected dephasing noise is much stronger than the noise causing the energy relaxation.

2. This is not unique to our engineered-noise experiment. When used as a sensor of their environmental dephasing noise, superconducting qubits (and other tunable qubit modalities) are biased at a flux-sensitive point (dephasing-sensitive point), where the qubit is entirely dominated by dephasing noise. In fact, these bias points are fairly common (predominant); in contrast, the dephasing-insensitive “sweet spot” only exists over a rather restricted range of biases.
3. In general, the T_1 contribution can be measured independently in the lab frame, and then accounted for explicitly. Please see Supplementary Material 7 for details.

In response to the referee comment, we now have added the following sentence in section “Spin-locking Noise Spectroscopy in a Multi-level Sensor” (p. 2) to alert the reader to the supplementary material that addresses the impact of T_1 to the protocol.

“The impact of energy relaxation (T_1) on our protocol is discussed in Supplementary Material 6.”

Please also note that we did have (and retain) the following sentence in the last paragraph of Section 2.B Noise Spectroscopy Protocol (not a new sentence in this response):

“To validate the protocol, we will perform the spin relaxation experiments both in the presence and in the absence of engineered noise, and distinguish the contributions of T_1 decay and native dephasing noise from the estimation of $\tilde{S}^{\perp}_{\{j-1,j\}}(\omega)$ (see Supplementary Material 7 for details)”

[1] F. Yan et al. *Nature Comms.* **4**, 2337 (2013)

[2] U. von Luepke et al. *arXiv*, 1912:04892 (2019)

4) This work considers noise on two of the three driving mechanisms in the circuit (flux, readout). Speculating on what, if anything, could be said about shot noise on the “Control line” would be useful in mastering this type of QNS for system-level analysis.

We thank the referee for raising this interesting point.

The control line in Fig. 2 (we usually call this the ‘charge line’) is capacitively coupled to the transmon. Therefore, voltage fluctuations on this control line result in fluctuations of the offset charge in the transmon. As discussed in Ref. [1], the circuit Hamiltonian of the transmon is

$$\hat{H} = 4E_C(\hat{n} - n_g)^2 - E_J \cos \hat{\phi},$$

where n_g corresponds to the offset charge that can fluctuates due to noise on the control line.

Notably, using our protocol, this charge noise can *also* be distinguished from the other two dephasing noise mechanisms (flux and photon shot noise) discussed in the paper. We note in the discussion section of the paper that distinguishing more than two noise mechanisms would require measuring more than two noise spectra across different level transitions. For example, we could measure, at least, noise spectra for $|0\rangle\text{-}|1\rangle$, $|1\rangle\text{-}|2\rangle$, and $|2\rangle\text{-}|3\rangle$ transitions in order to distinguish the noise contributions from the three different noise mechanisms. Although we did not show this in the paper, we do discuss the possibility as a natural extension of the concept developed in this work.

[1] J. Koch *et al.*, *Phys. Rev. A* **76**, 042319 (2007)

5) On page 1, during the listing of qubit platforms that have tested QNS protocols, consider adding two canonical references [NMR: Vandersypen and Chuang RMP (2005); SC Qubits: Ithier et al PRB (2005); and another platform Rydberg atoms [Carter and Martin PRA (2013)] which uses spin locking.

We thank the referee for bringing relevant references to our attention.

- I. J. D. Carter and J. D. D. Martin, *Phys. Rev. A* **88**, 043429 (2004)
- II. L. M. K. Vandersypen and I. L. Chuang, *Reviews of Modern Physics* **76** (2004)
- III. Ithier *et al.* *Phys. Rev. B* **72**, 134519 (2005)

All of these references have been added (Refs. 6, 10, and 13) in the list of qubit platforms that have tested QNS protocols (p. 1). Reference II is a broad review on NMR based control pulse techniques, so we also cited this paper (Ref. 13) in Section 2. B. Noise Spectroscopy Protocol, where control pulses ($X(\pi)$, $Y(\frac{\pi}{2})$), for noise spectroscopy are introduced (p. 3).

Response to Reviewer #2

The authors present an experimental and theoretical study of an application of spin-locking protocol of noise spectroscopy to a transmon qubit. This qubit differs from many others due to its weak anharmonicity and the fact that the qubit is based on two out of many energy levels - but one can easily drive the state out of the "qubit subspace" by applying a driving field of a slightly different frequency than the one used to drive the Rabi oscillations within this subspace

The spin-locking protocol of noise spectroscopy was previously shown in Refs [13] and [29] by some of the Authors of the manuscript. The novel factor in this work is taking into account the multi-level energy spectrum of the transmon device. Its presence is both a nuisance (as one has to abandon the two-level system approximation, and everything gets complicated) with which the Authors deal in a transparent way, but, as the Authors show, it also brings benefits, as one can consider spin-locking within various two-level subspaces, and draw interesting conclusions from comparison of results of noise spectroscopy for distinct subspaces.

The paper describes work that pushes the noise spectroscopy with transmons (and related qubits) into an interesting direction, but I am not fully convinced if the obtained results are going to significantly influence the field, or if they are a slightly involved extension of [13] and [29], in which more realistic structure of the system is taken into account - this is mostly related to point 3 below.

We would like to thank the referee for the detailed and constructive comments. We have carefully reviewed the comments and have revised the manuscript accordingly. In the following, our responses are given in a point-by-point manner.

(1) In the abstract we can read that the discussed method "extends the spectral range" of noise spectroscopy - but the limitations on this range specific to the more widely used dynamical decoupling noise spectroscopy and on the other hand to the spin-locking method are not discussed in the paper, and no citation is given to another work in which this is investigated. This is a clear omission: the reader is not given a justification why one should rather use spin locking instead of e.g. dynamical decoupling to characterize dephasing noise (in the original Hamiltonian) for transmons.

We thank the referee for raising an important point. Indeed, dynamic decoupling (D.D.) noise spectroscopy during predominantly free evolution has been widely used to characterize dephasing noise in various qubit systems. However, even within a two-level approximation, there are advantages to using a spin-locking (driven evolution) approach to noise spectroscopy. For example,

- 1) D.D. noise spectroscopy uses multiple, ideally instantaneous control pulses, which are used to flip the qubit states in the time domain and thereby realize a desired filter function in the frequency domain. Although we often think of these as narrow filters – ideally delta functions – that can sample the noise at any particular frequency, in practice, the width of the filter is determined by both the number of pulses and the duration of the experiment. Narrow filters require large numbers of pulses, which are not instantaneous in practice and take up available free-evolution time, and therefore lead to longer experiments (fighting against coherence times). In turn, high-frequency spectroscopy requires small time separation between pulses, yet high-fidelity pulses are not boxcars, but generally have Gaussian or cosine envelopes, requiring a minimum time between pulses to remain accurate. Taken together, this trade space can somewhat contradict and may make high-frequency noise spectroscopy with high precision a challenge. Spin locking does not have these issues, because it is essentially a single quasi-continuous drive.
- 2) In addition, practical D.D filters have a rather broad bandwidth (compared with spin locking), and a deconvolution step is required to extract the noise about some frequency. Furthermore, in many common approaches like the CPMG sequence, the filter function has one predominant peak and smaller lobes away from this peak, which exacerbates the deconvolution problem (a notable exception is the use of Slepian pulses, but these still lead to relatively broad filters). In contrast, the spin-locking approach uses a single, continuous drive, and we simply monitor decoherence times within the driven qubit basis to extract noise at the Rabi frequency with relatively small "filter bandwidth" (limited by coherence). This approach is much simpler to implement and generally more precise.
- 3) In practice, the DD control pulses are generally imperfect [1, 2]. These imperfections in control pulses (= control errors) limit the spectral range of the sensor, as discussed in Ref. [3]. Namely, the spectral range is mainly limited by control imperfections, rather than physical constraints inherent in the system. In contrast, the spin-locking noise spectroscopy does not require fast-pulsed control (it is a single quasi-CW drive); therefore, it can characterize higher frequency noise than the D.D. based noise spectroscopy as shown in Ref. [4] (this is also related to point #1 above).
- 4) In quantum systems, noise can affect systems both classically and non-classically. In the case of non-classical (quantum) noise, the noise seen by the qubit generally features an asymmetric power spectrum ($S(\omega) \neq S(-\omega)$). In general, the D.D. noise spectroscopy extracts the symmetric part of the noise PSD, $S_{\text{eff}}(\omega) = S(\omega) + S(-\omega)$, so it does not distinguish

between classical and non-classical noise processes [5], as opposed to the spin-locking based noise spectroscopy [6]. Since distinguishing classical and non-classical noise is of both fundamental and practical significance for quantum metrology and bath engineering [7], we believe this is another advantage of spin locking.

Nevertheless, we do agree that extending D. D. noise spectroscopy to a multi-level sensor would be an interesting avenue for future research. Notably, the idea of discriminating noise sources using a multi-level system is immediately applicable to D.D. based QNS protocols.

In response to the referee's comment, we have added the following sentence in the discussion.

“Although we mainly focus on the spin-locking based multi-level QNS throughout this work, extending the dynamic decoupling based noise spectroscopy protocols [cite{Alvarez et al (2011), Paz-Silva et al. (2014), Norris et al. (2016)}] to multi-level systems would also yield improved QNS performance. Notably, the idea of discriminating noise sources by employing multiple level transitions as distinct spectrometers is immediately applicable to dynamic decoupling based approaches. In view of recent advances in optimal band-limited control [cite{Norris et al (2018), V. Frey et al. (2017)}], we expect the implementation of dynamic decoupling based QNS using multi-level sensors will augment knowledge about noise sources in a manner similar to the spin-locking approach described here.”

- [1] L. Cywinski *et al. Phys Rev. B* **77**, 174509 (2008)
- [2] J. Bylander *et al. Nature Phys.* **7**, 565-570 (2011)
- [3] G. Alvarez and D. Suter, *Phys. Rev. Lett.* **107**, 230501 (2011)
- [4] F. Yan *et al. Nature Comms.* **4**, 2337 (2013)
- [5] L. Norris *et al. Phys. Rev. Lett.* **116**, 150503 (2016)
- [6] F. Yan *et al. Phys. Rev. Lett.* **120**, 260504 (2018)
- [7] K. W. Murch *et al. Phys. Rev. Lett.* **109**, 183602 (2012)

2) The issue of what terms couple longitudinally/transversely in the original Hamiltonian / in the Hamiltonian using the basis of dressed states, and their relation to longitudinal/transverse relaxation could be perceived as confusing by the reader. First, while in the appendix it is clearly stated that "longitudinal noise in the lab frame for a multi-level system leads to both transverse and longitudinal noise in the spin-locking frame", such a simple statement seems to be missing from the main text, where it would be quite useful. Also, while it is known that longitudinally coupled noise causes transverse relaxation (and transversely coupled noise causes longitudinal relaxation), having both the terminology such as the one below Eq (1): "the $B_{(j)}(t)$ correspond to the time-dependent noise operators that longitudinally couple to the j -th level of the sensor", and then "the emergence of non-zero transverse relaxation $B_{\parallel(j-1;j)}(t)$, terms" is confusing. It would be better to call $B_{\parallel(j-1;j)}(t)$ longitudinal (in the spin-locked frame) noise that causes transverse relaxation.

We thank the referee for the suggestion to define the terms in a clearer way. Following the suggestion, we have added the sentence below Eq. (3), which introduced the terms $B_{\parallel(j-1,j)}(t)$ and $B_{\perp(j-1,j)}(t)$ clearly as follows:

“The longitudinal noise in the lab frame (Eq.1) for a multi-level system leads to both transverse and longitudinal noise in the spin-locking frame. As a result, the longitudinal noise operator B in the lab frame is transformed into the spin-locking frame as a transverse noise operator $\tilde{B}^{(j-1,j)}$ ”

$1, j)_{\perp}(t)$, which leads to longitudinal relaxation, and the longitudinal noise operator $\tilde{B}^{(j-1, j)}_{\parallel}(t)$, which leads to transverse relaxation, within the j -th spin locking subspace.”

3) Why only the influence of engineered noise was investigated with the proposed method? Isn't there anything to say about the intrinsic noise? After the spectrometry protocol is validated for a known injected noise spectrum, it could be used for additional characterization of the intrinsic noise affecting the qubit. In the appendix it is discussed how the influence of that noise is subtracted from results to reconstruct the engineered noise. But why not characterize the intrinsic noise with the discussed method? At least a comment would be welcome, and showing some result would make the paper much stronger.

This question is related to a more general issue: the transmon was designed in order to limit the amount of dephasing that the qubit experiences, with the price paid by decreasing the anharmonicity of the system spectrum. Now, in order to sense the longitudinal (in the original frame) noise, one has to do spin locking and consider the multi-level structure of the system. But is the dephasing noise affecting the qubit strong enough/interesting enough to go through all this effort to characterize it?

We thank the referee for the important comment.

As pointed out by the referee, the transmon was designed to suppress dephasing due to charge noise at the expense of energy level anharmonicity. Our detector is not sensitive enough to measure the residual charge noise dephasing of the transmon. As for flux noise, our detector is generally sufficiently sensitive to detect $1/f$ flux noise through the transmon qubit loop away from its first-order flux-insensitive point. However, we intentionally minimize the impact of intrinsic $1/f$ flux noise to the sensor by carefully choosing its operating point (flux bias) and operating frequency range (measured spectral range), such that the sensor is dominated by engineered noise. This was clearly a choice. In practice, when used as a sensor of flux noise, one would of course bias the qubit at a flux-noise-sensitive point.

To address the referee comment, we show that the qubit is sensitive to intrinsic flux noise depending on its bias point. We numerically estimate the spin relaxation rate $\Gamma_{1\rho}$ of the qubit due to its intrinsic $1/f$ flux noise. **Figure 1(a)** shows the qubit flux sensitivity $d\omega_{s}^{(0,1)}/d\Phi_{\text{ext}}$ as a function of flux bias. As expected the flux sensitivity increases as the flux bias approaches $0.5 \Phi_0$. By plugging the literature number of flux noise amplitude ($1 \mu \Phi_0/\sqrt{\text{Hz}}$) [1], we estimate the spin relaxation rates due to $1/f$ flux noise at Rabi frequencies = 1kHz and 1MHz (**Figure 1(b)**). Then, we compare these two numbers with the spin relaxation rate $1/2T_1$ due to T_1 process. (Please see **Supplementary Material 6**, which we now have added to discuss the impact of T_1 on our protocol). If the dephasing rate is much smaller than $1/2T_1$, the sensor may not be able to detect the dephasing noise reliably. Conversely, if the dephasing rate is comparable to (or faster than) $1/2T_1$, then the sensor can measure the dephasing noise (= $1/f$ flux noise, here) reliably. As shown in Figure 1(b), the spin relaxation rate at Rabi Freq = 1kHz due to $1/f$ flux noise is much higher than the T_1 contribution, which means that we can measure the intrinsic flux noise reliably at Rabi frequency ~ 1 kHz. If we bias close to $0.5 \Phi_0$, we can measure intrinsic flux noise out to almost 1 MHz. To reach higher frequencies, one would need to design the transmon with a larger circulating current, or one could also move to a flux qubit. For diagnostics of a fabrication process or environmental sensing, these are entirely reasonable approaches.

And, as qubit T1 improves, in general, qubits will be better able to resolve 1/f noise at higher and higher frequencies.

Figure 1. (a) Flux sensitivity of the 0-1 transition frequency $\omega_{s^{(0,1)}}$ as function of flux bias. (b) Spin relaxation rate due to 1/f flux noise (blue and orange) and T1 process (red). We assume the literature number of flux noise amplitude A (the noise PSD is given by $S(\omega) = A/\omega$, where $\sqrt{A} = 1e-6 \Phi_0/\sqrt{\text{Hz}}$)[1] The black dashed vertical lines correspond to flux bias points, where we biased the sensor.

Along these lines, although we employ the transmon as a multi-level noise sensor, our methodology is portable to other anharmonic multi-level systems, such as the C-shunt flux qubit [1] and the fluxonium [2], which generally experience strong dephasing noise at certain bias points. Since the sensitivity of the qubit energies to various noise sources differs by qubit designs, employing other superconducting qubit variants as multi-level noise sensors will enable us to explore various noise sources, an interesting avenue for future research. That being said, we emphasize that our transmon sensor is a test vehicle to validate the new approach to distinguish and identify noise sources by exploiting multiple energy level structure of a sensor.

In response to the referee's comment, we now have added the following paragraph in the discussion.

In this paper, we demonstrated our protocol by measuring engineered noise in the flux-tunable transmon sensor. We chose the operating point (flux bias) and operating frequency range (measured spectral range) of the sensor, such that it is dominantly affected by the engineered noise. However, the technique discussed here can be also applied to measure intrinsic noise of transmons such as 1/f flux noise~\cite{Oliver2013, Paladino2014, Braumuller2020}. Notably, by biasing the sensor at more flux-sensitive point, the sensitivity to flux noise can be further increased in order to detect intrinsic flux noise.

While we employ a flux-tunable transmon as a multi-level noise sensor, our methodology is portable to other anharmonic multi-level systems, such as the C-shunt flux qubit~\cite{Yan2016} and the fluxonium~\cite{Manucharyan2009, Nguyen2019}. Since the sensitivity of the qubit energies to various noise sources differ by qubit design, employing other superconducting qubits as multi-level noise sensors will enable us to explore various noise sources. We also envision the spin-locking QNS protocols

– whether in a TLS approximation or a multi-level system – being used for other qubit modalities, such as quantum dot qubits or trapped ion qubits, as sensors of their local environments, such as their substrates or surface traps.

As detailed in Supplementary Material 6, the T1 of the qubit can limit its noise sensitivity. However, as T1 is improved through a combination of qubit design^{\cite{Nguyen2019}} and advanced materials^{\cite{Place2020}}, the sensitivity and utility of our approach also improves. Using diagnostic techniques such as the QNS protocol developed here to identify and characterize noise sources is an important step towards mitigating and eliminating them.”

[1] F. Yoshihara et al. *Phys Rev. Lett.* **97** 167001(2006)

[2] F. Yan et al. *Nature Communications* **7** 12303 (2016)

[3] V. Manucharyan et al. *Science* **326**, 113-116 (2009)

Response to Reviewer #3

Multi-level quantum noise spectroscopy describes a method of detecting and characterizing noise using a multi-level quantum sensor, using a superconducting flux qubit for the experimental demonstration. This is an extension of previous work by Oliver et. al., primarily reference [13], which used spin locking experiments to measure noise in the rotating frame. This paper proposes a method for using multi-level systems (such as a weakly anharmonic flux qubit) for noise spectroscopy. This requires a complete analysis that accounts for the higher levels on the rotating frame of any two driven levels. It also allows the experimentalist to distinguish between different types of noise by probing higher levels.

The essential experiment is to first bring the system to the starting state by a series of pi-pulses and then perform a Ramsey experiment around the application of a spin locking field. Varying the length of the spin locking drive gives the polarization as a function of time and $T1\rho$. These quantities are enough to calculate the power spectral density for the pair of levels that correspond to the spin locking drive.

The authors demonstrate this method on a flux qubit by measuring the power spectral densities of the 0-1 and 1-2 subspaces. They use engineering flux noise and photon shot noise to validate the method and show that they can distinguish between the two types of noise.

Overall, this work is well presented and clearly demonstrates a new method of noise spectroscopy. The careful theoretical analysis in the supplementary materials is the strength of the paper. The analysis of the effect of higher levels on the rotating frame Hamiltonian is technically sound and the additional description of pump-probe is a useful resource.

I recommend the paper for publication with a few comments:

We thank the referee for the positive assessment and valuable comments. We have carefully considered all the comments and suggestions and modified both the manuscript and supplement accordingly. Please see below for our point-by-point responses.

One concern is that the approach is tailored to flux qubits and the particular kind of noise that may be present in flux qubits (photon shot noise, flux noise). I would suggest adding more commentary on how this methodology could be used in other systems (the authors do mention charge noise using additional higher levels).

We thank the referee for raising this question of generalizing our method. To clarify, our experiment utilizes a flux-tunable transmon as a multi-level noise sensor. Our methodology is portable to other quantum systems with an anharmonic multi-level energy structure. For example, variants of superconducting qubits such as the C-shunt flux qubit [1] and the fluxonium [2] are candidates for multi-level noise sensors, since they have multiple level transitions which are addressable by microwave tones. Thus, our protocol is not limited to transmons or to transmon-specific noise channels. It is worth noting that different qubit modalities may have different levels of sensitivity to different types of noise.

To clarify these points in the manuscript, we now have added the following paragraph in the discussion.

“While we employ a flux-tunable transmon as a multi-level noise sensor, our methodology is portable to other anharmonic multi-level systems, such as the C-shunt flux qubit^{~\cite{Yan2016}} and the fluxonium^{~\cite{Manucharyan2009, Nguyen2019}}. Since the sensitivity of the qubit energies to various noise sources differ by qubit design, employing other superconducting qubits as multi-level noise sensors will enable us to explore various noise sources. We also envision the spin-locking QNS protocols – whether in a TLS approximation or a multi-level system – being used for other qubit modalities, such as quantum dot qubits or trapped ion qubits, as sensors of their local environments, such as their substrates or surface traps.”

[1] F. Yan *et al. Nature Comms.* **7**, 12694 (2016)

[2] Long B. Nguyen *et al. Phys. Rev. X.* **9**, 041041 (2019)

Additionally, the system used was a qubit that was fairly isolated from other coupled qubits. How would stronger coupling impact the analysis?

We thank the referee for raising this interesting point. In the following, we describe how the analysis is affected for strongly coupled qubits. We first categorize the case into two possible scenarios:

- (1) The coupling strength g between qubits is smaller than the qubit-qubit frequency detuning ($g \ll \text{freq. detuning}$).
- (2) The coupling strength g is comparable to the frequency detuning ($g \geq \text{freq. detuning}$)

Case (1): If the spin locking Rabi frequency Ω is smaller than the qubit-qubit frequency detuning, the additional level dressing resulting from the qubit-qubit interaction can be neglected and the analysis presented in our manuscript remains valid.

Case (2): the qubit sensor can no longer be assumed as a separate system since the coupling to another qubit is substantial. Here, we must consider the effect of the level dressing (or interaction) between two coupled qubits. Although this is not within the scope of our paper, we believe that extending our protocol to coupled qubit systems will be an interesting avenue for future research. This may be an interesting

direction for investigating correlations in specific dephasing noise channels. We note that the joint noise spectra for a two-qubit system has recently been demonstrated using the conventional (two-level) spin locking based QNS [1].

[1] U. von Luepke et al. *arXiv*, 1912:04892 (2019)

The results with engineered noise are convincing, but it would be worth adding some discussion on the sensitivity of this method.

We thank the referee for making this important point.

Coherence is one limit of the qubit sensitivity. We have added Supplementary Material 6 to discuss the impact of T1 on the noise sensitivity of our protocol. As detailed in Supplementary Material 6, if the energy relaxation rate ($1/T_1$) of the sensor is much faster than the dephasing rate (due to the intrinsic dephasing noise), it can mask its contribution and limit sensitivity.

In response to the referee's suggestion, we have added the following sentence in the section Experimental Validation (1st paragraph in p.8) to point the reader where we discuss the noise sensitivity due to T1.

“Lastly, at 20 MHz, above which no external noise was applied, the data exhibit a discrete jump down to the sensitivity limit of the experiment (See Supplementary Material 6 for discussion on the sensitivity limit due to T_1 of the sensor).”

It's not clear in the pump-probe experiment (Fig. S5) and the Rabi experiment (Fig 3(a)), if A_{drive} is an extrapolation from the low frequency measurements assuming a two-level system or if there a fit parameter to the multi-level model to extract A_{drive} . Since the deviation of the Rabi frequency from A_{drive} is important for this work, I think it is worth being explicit about how A_{drive} is determined.

This is indeed an important point, on which we had been insufficiently explicit. We determine A_{drive} by assuming a linear dependence of the Rabi frequency in the weak drive limit (Rabi frequency < 5 MHz). To clarify this point, we now have added the following sentence in the 2nd paragraph in page 6 (Section 3. Experimental Validation),

“To determine A_{drive} , we assume a linear dependence in the weak driving limit, where Rabi frequency < 5 MHz. From this linear dependence, we could extrapolate A_{drive} to the strong driving regime.”

In Fig. 3(d) it is difficult to tell the marker shapes apart, perhaps another shape or different color would make this plot clearer.

We thank the referee for pointing out the unclear markers in Fig. 3(d). In response to this comment, we now have changed the marker shapes of $SL^{\{1,2\}}$ from crosses 'x' to hollow circles 'o' to make it more distinguishable from $SL^{\{0,1\}}$ data.

Fig. 3(d) previous version: using 'x'-shaped markers

Fig. 3(d) current version: using hollow 'o'-shaped markers

REVIEWERS' COMMENTS

Reviewer #1 (Remarks to the Author):

The authors have suitably addressed the points raised by the the referees. I recommend the manuscript for publication in Nature Communications.

Reviewer #2 (Remarks to the Author):

While I think it would be better if the discussion comparing the DD and spin-locking approaches that is contained in the response to my remark (1) appeared also in the manuscript, after reading the paper and responses I can recommend publication in Nature Communications.

By the way, "recent advances in optimal band-limited control" mentioned in the new version of the manuscript can be applied only to the situation in which the noise that one wants to characterize is in fact a multiplicative noise disturbing the amplitude of Rabi driving. They cannot be applied to the more commonly encountered case of noise affecting additively the qubit energy splitting. But maybe the spin-locking kinds of protocols are a good avenue to make some of those ideas.

Reviewer #3 (Remarks to the Author):

The authors have addressed all of my comments, and additional changes made in response to the other reviewers have improved the paper. I am happy to recommend publication.

Re: NCOMMS-20-11832A

In the following, **blue** indicates referee text, **black** indicates our response to the referee. All the changes we implemented in the main text and supplement are clearly marked in **red**.

Response to Reviewer #2

While I think it would be better if the discussion comparing the DD and spin-locking approaches that is contained in the response to my remark (1) appeared also in the manuscript, after reading the paper and responses I can recommend publication in Nature Communications.

We thank the referee for the suggestion of adding the discussion contained in our previous response.

Following the referee's suggestion, we have added **Supplementary Note 8. Advantages of the spin-locking QNS over the dynamic decoupling QNS** to discuss advantages of the spin-locking based noise spectroscopy protocols, and thus justify why we focus on the spin-locking approaches throughout this work. We also have added the following sentence in the conclusion section (p. 8) to point the reader where we discuss the advantages of the spin-locking approaches.

"Although we mainly focus on the spin-locking based multi-level QNS throughout this work, extending the dynamic decoupling (D.D.) based noise spectroscopy protocols to multi-level systems would also yield improved QNS performance (see **Supplementary Note 8 for a discussion of why we focus on the spin-locking based approaches rather than the D.D. based approaches throughout this work**)."

By the way, "recent advances in optimal band-limited control" mentioned in the new version of the manuscript can be applied only to the situation in which the noise that one wants to characterize is in fact a multiplicative noise disturbing the amplitude of Rabi driving. They cannot be applied to the more commonly encountered case of noise affecting additively the qubit energy splitting. But maybe the spin-locking kinds of protocols are a good avenue to make some of those ideas.

We thank the referee for raising an important point about the Refs. [1,2] that we cited in the new version of the manuscript.

As the referee pointed out, the optimal band-limited control protocol introduced in Refs. [1,2] has a limitation. The control variable (= the amplitude of the applied drive) couples to additive dephasing in a highly nonlinear way, which makes it difficult to extract the spectrum of the dephasing noise from the sensor's response (i.e. the filter functions for dephasing sensing are not spectrally concentrated. please see the section IV in p. 21 of Ref. [2] for details). Because of the nonlinearity, they could not characterize dephasing by using their "plain" DPSS modulation based protocol.

However, we recently noticed that the nonlinearity issue can be circumvented by using a more complicated kind of DPSS modulation [3]. Ref. [3] shows that the sensor's response to dephasing noise can be linearized by using a finite-difference modulation scheme, as explained on pp. 5-6 of Ref. [3]. This scheme gives spectral concentration in the filter functions for both dephasing and driving amplitude noise and thus enables characterization of both types of noise without spectral leakage [3].

To reflect the comment from the referee, we have deleted the citations [1,2] from "recent advances in optimal band-limited control", since the protocol introduced in Refs. [1,2] cannot be applied to measure

the spectrum of dephasing noise (which we focus on throughout this work). Instead, we have cited Ref. [3] that introduces the optimal band-limited control protocol that can be used to characterize dephasing without spectral leakage.

[1] V. M. Frey, S. Mavadia, L. M. Norris, W. de Ferranti, D. Lucarelli, L. Viola, and M. J. Biercuk, *Nature Communications* **8**, 2189 (2017).

[2] L. M. Norris, D. Lucarelli, V. M. Frey, S. Mavadia, M. J. Biercuk, and L. Viola, *Phys. Rev. A* **98**, 032315 (2018).

[3] V. Frey, L. M. Norris, L. Viola, and M. J. Biercuk, *Phys. Rev. Applied* **14**, 024021 (2020).